# CNA Profiling of Single CTCs in Locally Advanced Esophageal Cancer Patients during Therapy Highlights Unexplored Molecular Pathways

**DOI:** 10.3390/cancers13246369

**Published:** 2021-12-19

**Authors:** Giulia Gallerani, Tania Rossi, Martina Valgiusti, Davide Angeli, Pietro Fici, Sara De Fanti, Erika Bandini, Claudia Cocchi, Giovanni Luca Frassineti, Massimiliano Bonafè, Francesco Fabbri

**Affiliations:** 1Biosciences Laboratory, IRCCS Istituto Romagnolo per lo Studio dei Tumori (IRST) “Dino Amadori”, 47014 Meldola, Italy; tania.rossi@irst.emr.it (T.R.); pietrofici@gmail.com (P.F.); erika.bandini@irst.emr.it (E.B.); claudia.cocchi@irst.emr.it (C.C.); francesco.fabbri@irst.emr.it (F.F.); 2Department of Medical Oncology, IRCCS Istituto Romagnolo per lo Studio dei Tumori (IRST) “Dino Amadori”, 47014 Meldola, Italy; martina.valgiusti@irst.emr.it (M.V.); luca.frassineti@irst.emr.it (G.L.F.); 3Unit of Biostatistics and Clinical Trials, IRCCS Istituto Romagnolo per lo Studio dei Tumori (IRST) “Dino Amadori”, 47014 Meldola, Italy; davide.angeli@irst.emr.it; 4Interdepartmental Centre “Alma Mater Research Institute on Global Challenges and Climate Change (Alma Climate)”, University of Bologna, 40126 Bologna, Italy; sara.defanti2@unibo.it; 5Department of Experimental and Diagnostic Medicine, University of Bologna, 40126 Bologna, Italy; massimiliano.bonafe@unibo.it

**Keywords:** circulating tumor cells, single cell analysis, CNA profiling, esophageal cancer

## Abstract

**Simple Summary:**

In the present work, we describe the evolution of circulating tumor cells (CTCs) released into the bloodstreams of 11 patients affected by locally advanced esophageal cancer (EC) during clinical treatments. We aimed at characterizing identified CTCs in depth both phenotypically as well as through single cell copy number aberrations profiles and to investigate the features of CTCs from relapsed patients, if present. We found that locally advanced EC spreads circulating tumor cells with both epithelial and mesenchymal phenotypes during the course of therapy. CTCs of relapsed patients display higher levels of genome disruption to those of disease-free patients. Specific enriched terms emerged from copy number aberration analysis of CTCs of relapsed patients.

**Abstract:**

Background: Here, we monitored the evolution of CTCs spread in 11 patients affected by locally advanced EC who were undergoing therapy. Methods: In this perspective study, we designed multiple blood biopsies from individual patients: before and after neoadjuvant chemo-radio therapy and after surgery. We developed a multi-target array, named Grab-all assay, to estimate CTCs for their epithelial (EpCAM/E-Cadherin/Cytokeratins) and mesenchymal/stem (N-Cadherin/CD44v6/ABCG2) phenotypes. Identified CTCs were isolated as single cells by DEPArray, subjected to whole genome amplification, and copy number aberration (CNA) profiles were determined. Through bioinformatic analysis, we assessed the genomic imbalance of single CTCs, investigated specific focal copy number changes previously reported in EC and aberrant pathways using enrichment analysis. Results: Longitudinal monitoring allowed the identification of CTCs in at least one time-point per patient. Through single cell CNA analysis, we revealed that CTCs showed significantly dynamic genomic imbalance during treatment. Individual CTCs from relapsed patients displayed a higher degree of genomic imbalance relative to disease-free patients’ groups. Genomic aberrations previously reported in EC occurred mostly in post-neoadjuvant therapy CTCs. In-depth analysis showed that networks enrichment in all time-point CTCs were inherent to innate immune system. Transcription/gene regulation, post-transcriptional and epigenetic modifications were uniquely affected in CTCs of relapsed patients. Conclusions: Our data add clues to the comprehension of the role of CTCs in EC aggressiveness: chromosomal aberrations on genes related to innate immune system behave as relevant to the onset of CTC-status, whilst pathways of transcription/gene regulation, post-transcriptional and epigenetic modifications seem linked to patients’ outcome.

## 1. Introduction

Esophageal cancer (EC) represents a significant burden on human health, being the sixth most common cause of cancer-associated death globally [1,2] with a 5-year survival rate that is less than 20%, including West and Asian countries [3]. Even for those patients (>50%) deemed fit for curative interventions with aggressive (despite non-targeted) neoadjuvant therapy followed by surgery, the outcomes are still poor, with a 5-year survival rate of <45% [4]. While EC treatments use a similar criterion, the underlying biological differences of esophageal tumor types are becoming evident [5]. Furthermore, in gut adenocarcinomas, widespread genomic disruption due to copy number alterations (CNA) differs across cancers and cancer types [6,7].

EC is notable among cancer types for its high degree of chromosomal instability [8] as well as genomic disruption [6], which is due to CNA involving whole chromosomes or chromosome arms [7,8,9].

Unlike the mutational status, CNAs events are restricted to EC and are not present on its precursor lesion, Barrett’s esophagus [10]; this makes CNA analysis highly suitable for the early detection of esophageal cancer cells. Furthermore, CNAs were adopted by Noorani et al. [11] to suggest the “diaspora” theory for EC dissemination, their high-resolution perspective study revealed that metastases directly resemble the primary site [11]. In this scenario, the analysis of circulating tumor cells (CTCs) represents a key player to understand EC aggressiveness related to tumor spreading. Previously, we and others illustrated the presence of CTCs in non-metastatic cancers, proving that tumor spreading starts in early cancer stages [12,13,14].

Esophageal CTCs in the blood of non-metastatic patients were found during multimodal treatment [15] and their presence was correlated with poor clinical outcome on patients that underwent esophagectomy solely [13,16].

In-depth CTCs analysis of different cancer types, such as breast and lung cancers, provided new insights about tumor spreading as well as identifying CNA signatures on relapsed patients [17]. It is therefore mandatory to investigate CTCs in EC too.

To better understand the role of CTCs in locally advanced non-metastatic EC, we designed a perspective study that necessitated multiple blood biopsies from individual patients drawn at different time points during treatment (neoadjuvant chemo-radiotherapy followed by surgery) (Appendix A).

Here, we set up a multi-antibodies analysis named Grab-all assay to assess CTCs for the epithelial (EpCAM [18]/E-Cadherin [19,20]/Cytokeratins [21]) and mesenchymal/stem (N-Cadherin [22,23]/CD44v6 [24,25,26,27]/ABCG2 [28,29]) cell phenotypes. All identified CTCs were sorted as single cells, subjected to whole genome amplifications and low-resolution whole genome sequencing to obtain the CNAs of individual CTCs (Appendix A).

Our data indicate that the Grab-all assay improves the identification of CTCs in peripheral blood of non-metastatic EC patients. CTC CNA profiling at the single-cell level in combination with a longitudinal study design highlights differences in the extent of genomic imbalance between CTCs isolated from relapsed patients as opposed to those without evidence of disease, as well as in relation to the time when the liquid biopsy was taken.

## 2. Materials and Methods

### 2.1. Patients Recruitment and Sample Collection

The study had been prospectively approved by the Research Ethics Committee from IRCCS Istituto Romagnolo per lo Studio dei Tumori (IRST) “Dino Amadori” (Italy) (reference, IRSTB017) according to the recommendations of the declaration of Helsinki, patient consented to the project were enrolled at IRCCS-IRST between 2013 and 2017.

Eligible patients were newly diagnosed with metastatic or locally advanced EC with treatment protocols of first line chemotherapy and neoadjuvant chemo-radiotherapy followed by surgery, respectively. Patients were staged, and managed standard treatment protocols were in accordance with international guidelines [30]. Clinical data including age, tumor staging, treatments protocols and clinical outcomes were collected. Disease-free patients were defined as those who resulted negative for tumor detection at CT scan after surgery for at least five years.

Metastatic patients’ blood was collected at study entry and at CT scan evaluations. Blood drawn of non-metastatic patients was collected before the administration of neoadjuvant therapy, after neoadjuvant therapy at CT-scan evaluation and a month after surgery concurrently with CT-scan. All blood drawn volumes intended for CTC analysis ranged between 18–20 mL; first 3 mL of blood were not included in CTC enrichment to avoid epithelial cells contamination.

### 2.2. Cell Culture

The MDA MB 436 breast cancer cell line and MCF10A epithelial breast cell line were purchased from ATCC. EC cell line OE33 was kindly donated by Prof. Nickolas Stoecklein (University Hospital Düsseldorf). MDA-MB 436 cell line was cultured in Leibovitz’s L-15 medium (GIBCO, Grand Island, NE, USA) with 10 ug/mL insulin (Sigma-Aldrich, Burlington, MA, USA), 100 U/mL penicillin-streptomycin (GIBCO, Grand Island, NE, USA), 2 mM glutamine (GIBCO, Grand Island, NE, USA) complemented with 10% FBS (GIBCO, Grand Island, NE, USA), in a non-free gas exchange condition.

MCF10A cells were cultured in DMEM/F-12, GlutaMAX™ medium (GIBCO, Grand Island, NE, USA) complemented with 5% Horse Serum (GIBCO, Grand Island, NE, USA) (GIBCO, Grand Island, NE, USA), 100 U/mL penicillin-streptomycin (GIBCO, Grand Island, NE, USA), 2 mM L glutamine (Invitrogen, Waltham, MA, USA), 100 mg/mL human epidermal growth factor (EGF) (Merck Millipore, Burlington, MA, USA), 1 mg/mL hydrocortisone (Sigma-Aldrich, USA), 1mg/mL cholera toxin (Sigma-Aldrich, Burlington, MA, USA) and 10 mg/mL insulin (Sigma-Aldrich, Burlington, MA, USA), and 100 U/mL penicillin-streptomycin (GIBCO, Grand Island, NE, USA) as mentioned [31].

OE33 cell line was cultured in RPMI 1640 medium (GIBCO, Grand Island, NE, USA) complemented with 10% FBS (GIBCO, Grand Island, NE, USA), and 100 U/mL penicillin-streptomycin (GIBCO, Grand Island, NE, USA).

All cell lines were maintained as a monolayer in a 37 °C incubator with 5% CO_2_ and sub-cultured at 70–75% confluence (approximately twice weekly).

We induced epithelial-to-mesenchymal transition (EMT) on MCF10A cell line by TGF beta treatment as reported [32]. Briefly, cells were treated with 10 ng/mL of TGF beta1 (Proprotech, London, UK) every 48 h for ten days. Morphological changes observation and images were performed by light microscopy Axiovert 200 (Zeiss, Oberkochen, Germany).

Grab-all assay and DEPArray analyses were performed on cells collected at four time points: before treatment, after 4, 6 and 10 days.

### 2.3. CTC Enrichment, Grab-All Assay and Single Cell Isolation

Blood samples (18–20 mL per sample) were collected in EDTA vacutainers (BD) and processed within 3 h. Label-independent enrichment of CTCs was carried out by density gradient centrifugation with OncoQuick^®^ as we previously used [33,34].

Next, Grab-all assay was performed on CTC samples. In details, samples were fixed in 2% paraformaldehyde (Miltenyi Biotech, Bergisch Gladbach, Germany) and incubated 30 min at 4 °C with cell membrane directed antibodies: EpCAM-PE (clone HEA-125) (Miltenyi Biotech, Bergisch Gladbach, Germany) 1:10; E-Cadherin-PE (clone 67A4) (Miltenyi Biotech, Bergisch Gladbach, Germany) 1:10; N-Cadherin AlexaFluor 647 (clone 8C11) (BD Pharmigen, Franklin Lakes, NJ, USA) 1:20; CD44v6-APC (clone 2F10) (R&D Systems, McKinley Pl NE, MN, USA) 1:20; ABCG2-APC (clone 5D3) (R&D Systems, McKinley Pl NE, MN, USA) 1:10; CD45-AlexaFluor 488 (clone GA90) (Life Technologies, Carlsbad, CA, USA) 1:10. We performed intracytoplasmic staining with pan-CKs-PE (clone C11) (Aczon, Bologna, Italy) 1:10, Hoechst 33,342 (1 μg/mL, Life Technologies, Carlsbad, CA, USA) diluted in a solution of 1× PBS/1% BSA (Sigma-Aldrich, Burlington, MA, USA) 0.3% Triton™ X-100 (Bio-rad, Hercules, CA, USA).

Stained samples were resuspended in 14 uL of SB115 Buffer (Menarini-Silicon Biosystems SpA, Bologna, Italy), loaded into DEPArray^TM^ chip (Menarini-Silicon Biosystems SpA, Bologna, Italy) and analyzed by DEPArray platform.

In order to avoid any technical bias related to acquisition of images, first we defined time exposure of DEPArray acquisition. We stained leukocytes only with CD45 antibody acquiring images of all fluorescent channels to define the maximum exposure time of FITC channel for saturation and to verify the absence of any autofluorescence signals on PE and APC channels. We followed the same approach to define epithelial and stem-mesenchymal tags (PE and APC channels, respectively) by staining a mix of MDA MB 436 and leukocytes cells. MDA MB 436 cells were stained with epithelial plus CD45 antibodies and subsequently with stem/mesenchymal plus CD45 antibodies. We configured time exposure of channels to prevent autofluorescence signals especially on FITC channel as well as saturation signals.

All acquired patient’s cell images were manually reviewed, cells classified as CTCs had nucleus rounded- or spindle- shaped morphology and were negative for CD45. Epithelial CTCs were positive only for epithelial tag (EpCAM/CKs/E-Cadherin), mesenchymal/stem CTCs were positive only for stem/mesenchymal tag (N-Cadherin/CD44v6/ABCG2) and bi-phenotipic CTCs when cells were positive for both tags. All routable CTCs were sorted as single cells, and single leukocytes were sorted as a reference diploid chromosome set for further molecular analysis. Cells were sorted into 0.2 mL tubes, and stocked at −80 °C after manual volume reduction and 1X PBS (GIBCO, Grand Island, NE, USA) wash until whole genome amplification (WGA).

### 2.4. Single CTC WGA and Quality Control (QC)

We performed WGA using Ampli1 WGA Kit (Menarini-Silicon Biosystems SpA, Bologna, Italy) according to manufacturer’s instructions except for the cell lysis step, which was conducted overnight into thermal cycler. DNA samples were stocked at −20 °C until use.

All the WGA products were quality checked adopting Ampli1™ QC Kit (Menarini-Silicon Biosystems SpA, Bologna, Italy) as per manufacturer’s instructions. Briefly, we performed a multiplex PCR to amplify four amplicons of different lengths. The amplification of the regions was assessed on a 2% agarose gel using Chemidoc instrument (Bio-rad, Hercules, CA, USA). This test was performed in order to check if amplification products were appropriate for further analysis. Lack of amplification means exclusion for library preparation and sequencing.

### 2.5. Library Preparation and Next Generation Sequencing (NGS)

DNA libraries for low-pass whole genome sequencing were generated using a streamline kit of WGA products named *Ampli1*™ LowPass Kit for Ion Torrent (Menarini-Silicon Biosystems SpA, Bologna, Italy) as reported [35,36,37]. WGA products of some samples were re-amplified by Ampli1™ ReAmp/ds following manifacturer’s protocol for the generation of appropriate libraries. Concisely, 10 µL of WGA products were linked to barcoded adaptors for Ion Torrent System by a PCR amplification. Next, libraries were quantified by Qubit™ 3.0 Fluorometer (Invitrogen, Waltham, MA, USA) and fragment length was assessed using Bioanalyzer High Sensitivity DNA Kit (Agilent). We combined libraries in an equimolar pool, and selected fragments of the pooled libraries ranging between 300 bp to 450 bp using E-Gel™ SizeSelect™ II Agarose Gels, 2% (Invitrogen, Waltham, MA, USA) on E-Gel™ iBase™ and E-Gel™ Safe Imager™ (Invitrogen, Waltham, MA, USA).

Whole genome sequencing of single CTCs and matched leukocytes was carried out on Ion PGM™ and Ion GeneStudio™ S5 System (Thermo Fisher, Waltham, MA, USA), loading 8-pooled samples on 318 and 520 chip, respectively. Sequencing was conducted following the guidelines reported in the protocol (emulsion PCR for 400-base-length libraries and 525-flows sequencing). For sequencing experiments conducted on Ion S5, chip preparation and emulsion PCR were performed on the Ion Chef instrument, using Ion 510™, Ion 520™ and Ion 530™ Kit reagents (Thermo Fisher, Waltham, MA, USA).

### 2.6. CNA Calling from Low-Resolution Whole Genome Sequencing

The output BAM files were given as input at Control-FREEC for CNA calling. We used two different non-parametric tests (Mann–Whitney and Kolmogorov–Smirnov) to assess the statistical significance of each putative CNA. The whole genome has been subdivided in windows (bin), whose size depends on the quality of the sequencing. Then, we generated plots showing copy numbers mapped for each bin, highlighting the points which are included in CNAs with a significance lower than 0.05 in both the non-parametric tests we applied.

### 2.7. Generation of Jaccard Indexes and Statistical Analysis

We computed a Jaccard Index (JI) for the detection of similarity, for what concerns CNA content, among each CTC against five leucocytes, which have been analyzed with the same workflow. JI has been computed by applying the following formula:$$JI\left(A,B\right)=\frac{A\cap B}{A\cup B}$$
in which *A* is the set of CNAs present in the CTC sample and *B* is the set of CNAs in a leucocyte. CTC samples have been grouped for access time, response and both of these conditions. Then, we assessed the presence of statistical differences among the groups. When comparing more than two CTC groups (i.e., access time and access time with response), we applied the Kruskal–Wallis test. Moreover, for each group pair in each condition, we applied the Mann–Whitney *U* test.

### 2.8. Identification of Recurrent Esophageal Cancer Aberrations on Single CTCs

Starting from a list of cytogenetic bands which were of specific interest for the disease [6], we extracted the number of CTCs with at least a CNA in the specific region, with a further discrimination for access time and patient response. A similar procedure has been applied to a list of genes included in the same cytogenetic bands.

### 2.9. Enrichment and Network Generation

For each sample, we extracted the list of genes located inside a significant CNA detected by the previously described pipeline. From this list, using R package *EnrichR* [38], we performed an enrichment analysis exploring GO Biological Process dataset and we filtered out terms with adjusted *p*-value higher than 0.05. Finally, for each dataset, we generated networks using R package *igraph* [39]. Inside these networks, nodes are the enriched terms while edges give the information about the presence of more than 10 shared genes among the two linked nodes. Moreover, the size of each node is related to the number of genes, while the sections of the pie chart inside nodes are proportional to the percentage of samples of access time.

## 3. Results

### 3.1. Grab-All Assay Setup

To examine CTCs phenotypes, we combined the Oncoquick marker-independent CTC enrichment [14,34] with the novel Grab-all assay. This procedure uses an array of antibodies against mesenchymal and epithelial antigens, which come under an Epithelial Tag (E-tag consisting of EpCam, E-cadherin, Keratins 4, 5, 6, 8, 10, 13, 18) or a Stem-Mesenchymal Tag (SM-tag consisting of N-Cadherin, CD44v6, ABCG2), as well as CD45 as a leukocyte marker. Our Grab-all assay was validated on the cell line MDA-MB436 [40,41], which expresses both epithelial and mesenchymal markers (Appendix A). Grab all assay sensitivity was also evaluated in MCF10A cells undergoing TGF-beta1-induced epithelial mesenchymal transition (EMT). In particular, these epithelial mammary cells are devoid of any SM-tag positivity in standard culture conditions, but after 4 days of TGF-beta1 exposure, when morphological changes of EMT began to appear, a decrease of the expression of E-tag became overt (Appendix A). Moreover, after six days of TGF-beta1 treatment, SM-tag weakly positive cells can be detected, with a continuous increase up to day 10 of treatment. Intermediate cells, with dual epithelial and mesenchymal phenotypes, as well as cells that are negative for both the E-tag and SM-tag, were all present at late stages of EMT.

### 3.2. Detection Efficiency of Grab-All Assay on Metastatic Esophageal Cancer Patients

First, we intended to verify the performance of the Grab-all assay on blood of patients with a high disease burden (primary tumor and multiple metastases, both unresected). Purposely, we applied the Grab-all assay on peripheral blood samples from four metastatic esophageal cancer patients undergoing clinical treatment (Appendix A). CTCs were detected in all blood samples (Appendix A and Figure 1) and the Grab-all assay identified CTCs expressing E-tag, or SM-tag or bi-phenotipic E-SM-tag.

### 3.3. Patient Characteristics

Then, the Grab-all assay was employed on peripheral blood samples from a total of 11 patients diagnosed with locally advanced non-metastatic esophageal cancer: 8 (72.7%) adenocarcinomas (EAC), and 3 (27.3%) squamous carcinomas (ESCC). Patient characteristics are summarized in Appendix A. All the enrolled patients were treated with neoadjuvant chemo-radiotherapy with surgery intent, detailed treatments are listed in Appendix A. Eight out of eleven patients (72.7%) underwent surgery. Three remaining patients (27.3%) did not undergo surgery due to disease progression. All the surgery patients showed a tumor reduction after neoadjuvant therapy as revealed by yp-stage [42] compared to c-stage [42]. Each patient sample was codified as follows: NMx (patient code) A/B/C (timepoint).

### 3.4. CTCs Identification in the Blood of Non-Metastatic Patients during Treatment

CTCs were detected in every patient (Table 1), albeit expressing distinct epithelial, mesenchymal and mixed epithelial/mesenchymal phenotypes, which were evenly distributed across the timepoints. Notably, CTCs from patient NM4 showed a change in phenotype before and after chemo-radiotherapy: all CTCs disclosed an epithelial phenotype before chemotherapy and turned out to show mixed epithelial/mesenchymal phenotype after treatment (Figure 2).

We found CTCs in six patients (54.5%) before they underwent neoadjuvant therapy (Figure 3), but we could no longer detect CTCs in three of them (NM5, NM7, NM8) after therapy. The three above-mentioned patients underwent surgery, but patient NM7 displayed CTCs even after surgery, and then died due to relapse. Conversely, patients NM5 and NM8 maintained a CTC-negative status and remained disease-free. Another group of CTCs-positive patients before neoadjuvant therapy (NM3, NM4, and NM16) maintained CTC positivity after treatment, with the exception of NM3, who died of sepsis. Despite surgery, patients NM4 and NM16 relapsed and died.

Concerning the five patients (NM1, NM2, NM6, NM9, NM10) who were CTCs-negative at diagnosis, we detected CTCs after neoadjuvant therapy in four of them (NM1, NM6, NM9, NM10). With the exception of NM1, who died after neoadjuvant therapy, the remaining three patients underwent surgery, but NM6 died before the third blood biopsy following neck metastases occurrence. Unexpectedly, we detected CTCs in patients NM2 and NM10 one month after surgery, but these patients remain disease free. Finally, NM9, who was CTC- negative, had a disease relapse after two years from surgery.

### 3.5. Copy Number Alterations in CTCs

Since CTC counts did not provide any information about clinical outcome, we opted for an in-depth single-cell analysis of CNAs of isolated CTCs. We first validated genomic analyses on OE33 esophageal cancer cell lines ex vivo spiked-in peripheral blood samples of healthy donors and stained with Grab-all assay. We sorted single and pooled OE33 cells using the DEPArray platform. Then, we performed whole genome amplification (WGA) on two single cells and a pool of four cells. Afterwards, samples were quality-tested to verify their genomic integrity. We generated NGS libraries and performed low-coverage whole genome sequencing as previously reported [35,36]. WGA amplification of the three samples resulted in clear aberrant CNA profiles (Appendix A). Clustering analysis between single and pooled cells showed high levels of concordance (>95%) (Appendix A). The genomic content of every DEPArray-sorted single CTC from patients was next assessed. All sorted CTCs (*n* = 57) conveyed an altered CNA profile, supporting further the validity of the Grab-all assay in identifying actual CTCs. In patient NM8, we identified CD45-negative cells without any specific tumor phenotype, subsequent CNA analysis revealed a normal diploid genome indicating that these cells were non-cancer related cells of unknown origin (Appendix A).

In order to compare and quantitatively assess the genomic imbalance of CTCs relative to a matched diploid genome, we calculated a Jaccard similarity index (JI) defined as the ratio of shared to all aberrations between single CTCs and leucocytes for each patient. JI values range between 0 and 1, with 0 indicating maximal genomic imbalance and 1 indicating complete overlap with a normal diploid genome [14].

CTC JIs were lower than 0.54, indicating that CTCs share less than 54% of CNA similarity with a normal diploid genome. The distribution of the CTC JIs across cell populations were visualized and compared using violin plots.

Then, we aimed at comparing the genomic imbalance of CTCs isolated from patients at different clinical stages. To this purpose, we plotted the distributions of CNA JIs of CTCs before neoadjuvant treatment, after neoadjuvant therapy and after surgery CTCs (Figure 4A). CTCs show significantly different values of genomic imbalance across time points (Kruskal–Wallis *p*-value = 3.1 × 10^−6^). After neoadjuvant therapy, CTCs showed a marked change in the profile and peak of the distribution, compatible with reduced genomic imbalance, compared to before treatment CTCs (Wilcoxon *p*-values of 4.2 × 10^−6^) and after surgery CTCs (Wilcoxon *p*-values of 0.00029). Conversely, genomic imbalance of before treatment CTCs and after surgery CTCs were not statistically different (Wilcoxon *p*-values of 0.59).

We then examined disease-free and relapsed patients CTCs (Figure 4B). Marked differences in genomic aberrations were found in the two CTCs populations (Kruskal–Wallis test *p* value = 1.3 × 10^−14^). A large proportion of relapsed patient CTCs show substantial genomic imbalance, with low dispersion of JI values (JI = 0.12) around the median value. At variance, disease-free patient CTCs convey low genomic imbalance and widely dispersed distribution of CTC JI values (JI = 0.19).

Finally, we investigated whether the difference between relapsed and disease-free patients emerged during treatment. We split the data from these two classes of patients in pre- and post- neoadjuvant therapy groups, which were found to be significantly different (Figure 4C). However, both before and after neoadjuvant therapy CTCs from patients who eventually relapsed were characterized by with higher genomic imbalance compared to CTCs from disease-free patients (Wilcoxon *p* value 5.4 × 10^−11^ and 2.2 × 10^−10^, respectively).

### 3.6. Investigation of Previously Reported Esophageal Cancer-Specific Chromosomal Aberration in CTCs

A set of specific chromosomal aberrations, known as focal copy number changes, have been previously reported in esophageal cancer [6]. We searched our CTC whole genome sequencing dataset for a list of 23 chromosomal amplifications and 19 chromosomal deletions (Figure 4D–G). The aim of this analysis was to characterize CTC’ CNAs in greater detail—against known EC-specific genomic aberrations—and to determine whether focal copy number changes were uneven across the groups (before neoadjuvant treatment, after neoadjuvant therapy and after surgery) or differed according to the outcome (disease-free versus relapse).

All the known chromosomal amplifications were detected in our dataset, while six out of 19 known deletions were absent (Figure 4D,F). Genes included in amplified and deleted regions are listed in Figure 4E,G, respectively.

All chromosomal amplifications were more frequent in CTCs from patients after neoadjuvant treatment, whereas their frequency was reduced in CTCs from patients before neoadjuvant treatment and even more in CTCs after surgery (Figure 4D). Unexpectedly, focal copy number amplifications were more abundant in CTCs isolated from patients who attained disease-free status, whereas CTCs from patients who eventually relapsed had very few amplifications.

Chromosomal deletions were more frequent in CTCs collected from patients before neoadjuvant therapy compared to CTCs after neoadjuvant therapy or surgery, being almost absent in these latter. Chromosomal deletions were also more frequently observed in CTCs from disease free patients than from relapsed patients (Figure 4F). Four chromosomal deletions, namely 5q11.2, 4q35.1, 7q36.3, and 2q33.3, were exclusively present in CTCs of relapsed patients.

### 3.7. Enrichment Analysis

Finally, to identify functional pathways of CNAs, we performed enrichment and network analysis.

According to Gene Ontology (GO) biological process domain, enriched terms network (adjusted *p* < 0.05) of all CTCs are illustrated in Figure 5. GO terms are represented as circles (nodes), in which percentage according to the timepoint of the CTCs is reported as a color (red, orange and yellow), the size of the nodes indicates the number of enriched CTCs, and edges (grey lines) represent at least ten common genes overlapping the two terms.

CTCs from all time points shared terms linked to the activation and modulation of innate immune system (yellow), as well as chemotaxis and migration of immune system cells (violet) and cellular response to cytokine stimuli (GO:0071345), thus setting up the main axis of the network. Strikingly, all the above-mentioned terms exhibited aberrant interferon-related genes (Appendix A) [14]. Sensory perception of bitter taste (pink) associated terms were also enriched in all timepoints, although disconnected from the main axis (Figure 5A).

Interestingly, the main axis of the network branches out with terms emerging only on CTCs identified post neoadjuvant therapy, in particular skin development (blue), nervous system development (GO:0007399) and modulation of the transcription and macromolecules biosynthesis (red and green). Of note, enriched terms of response to divalent ions are enriched mostly in after-surgery CTCs (Figure 5A).

We next sought to investigate differences between enriched networks of CNAs in relapsed patients compared to disease free patients CTCs (Figure 5B).

The main axis of the network is maintained, with six new emerged terms only in relapsed patients CTCs. In particular three terms (green) are located at the center of the network: positive regulation of transcription, DNA-templated (GO:0045893), positive regulation of transcription from RNA-polymerase II promotor (GO:0045944) and regulation of gene expression (GO:0010468). The other three terms did not share common genes: cytidine to uridine editing (GO:0016554), DNA cytosine deamination (GO:0070383) and DNA cytidine deamination (GO:0070383).

In Table 2, the mostly associated genes of these six terms that have been associated with esophageal cancer or other cancers are reported. In particular, these genes are associated with more aggressive tumor phenotype (SAL2), migration and invasion (TOX4, PRMT5, AJUBA) development and progression (ZNF219, CEBPE). TOX4 and APOBEC3 gene clusters have also been associated with genomic instability [43].

## 4. Discussion

In this study, we leveraged single cell analysis to investigate the features of CTCs from locally advanced EC patients during clinical management. The presence of these rare tumor cells involved in metastatic cascade have been demonstrated as valid prognostic marker of patients’ outcome [13], although the molecular characterization of CTCs is still lacking.

Our findings, albeit conducted on a small cohort, indicate that EC spreads epithelial as well as mesenchymal CTCs carrying high burden of genomic imbalance, that entails chromosomal aberrations in innate immune system-related *loci*. The identification and in-depth analysis of chromosomal aberrations in CTCs is the first step towards a more complete understanding of the genetic events that influence CTC’s metastatic potential [57], which will in turn improve our knowledge of EC metastatic cascade.

Through the application of the method we established, named Grab-all assay, we increased the number of identified CTCs with an epithelial phenotype: beside the well-established EpCAM [18] and CKs [21] we included E-cadherin [19,20] as a target. We also opted to include antibodies to identify CTCs with a stem/mesenchymal phenotype using molecular targets as N-Cadherin [22,23], CD44v6 [24,25,26,27] and ABCG2 [28,29] which allowed to identify both mesenchymal and mixed phenotype CTCs [58]. Overall, Grab-all assay maximized the number of CTCs for downstream analysis. Although most of the CTCs had an epithelial phenotype, the presence of CTCs with mixed phenotype and mesenchymal phenotypes strongly suggests the activation of EMT mechanisms in EC spreading [59]. Due to the small cohort of enrolled patients, it was not possible to associate total CTCs count or CTC phenotype to the clinical outcome. However, by exploiting whole genome copy number analysis on single CTCs, we noticed that the neoadjuvant therapy may has a selective action against CTCs, likely promoting a higher genomic imbalance. In fact, post-surgery CTC CNA is similar to that of before therapy CTCs as we observed in a previously study conducted on early breast cancer patients [14]. Considering genomic disruption levels of disease-free CTCs and relapsed patients CTCs, we observed a highly significant difference between the two groups. In particular, the latter CTCs displayed a higher level of CNA than the former, and it was present both before and after therapy. These findings lead us to hypothesize that EC aggressiveness may depend upon a CTC subpopulation with a highly aberrant chromosomal set up that is not affected by neoadjuvant therapy.

Considering that chromosomal amplification is an early event in esophageal cancer tumorigenesis [60] and our findings on CTC CNAs, we can therefore assume that chromosomal aberrations located in this sub-population of CTCs may confer tumor aggressiveness. Initially, we focused on the presence of previously reported esophageal-cancer specific chromosomal aberrations. Interestingly, all 23 chromosomal amplifications are represented, and 13 out of 16 deletions emerged mainly in post-neoadjuvant therapy CTCs. These findings suggest that neo-adjuvant therapy may trigger cellular mechanisms that lead to chromosomal amplification in regions where genes such as GATA6/4, MET, CDK6, MYB, PRKC1, MCL1, EGFR are present. These data confirm previous findings on the presence of GATA6 amplification after neoadjuvant therapy on primary tissue of EC [61]. Copy number abnormalities of MET [62], CDK6 and EGFR [63] were also observed in esophageal cancer tissues.

Remarkably, the enriched-network generated in all time points CTCs revealed aberrations in pathways related to the innate immune system. Strikingly, we previously identified similar findings in early breast cancer CTCs [14]. Hence, our data underpin a contribution of genome regions harboring genes of the innate immune system (interferon, in particular) in the acquisition of the CTC functional status [14]. In addition, network analysis showed that CTCs survived to neoadjuvant therapy had additional terms of nervous and epidermal system development and transcription modulation, suggesting that these genomic modifications may confer a selective advantage.

By exploring differences between CTCs of relapsed and disease-free patients, six pathways involving transcription/gene regulation, post-transcriptional and epigenetic modifications have emerged. This investigation revealed genes described in the literature, which play roles in esophageal cancer and other types of cancer as TOX4 and APOBEC3 gene cluster related to genomic instability, PRMT5, AJUBA promoting tumor migration and invasion, ZNF219, CEBPE and SAL2 genes associated to tumor progression and aggressiveness, respectively.

Our work demonstrates for the first time that CTCs of locally advanced EC may have a mixed phenotype with high levels of genomic disruption, particularly those CTCs found in relapsed patients. Furthermore, we have identified a set of aberrant genes common on relapsed patient CTCs that could play a role in the metastatic process and cancer recurrence.

Further experiments and analysis should be directed towards identified gene pathways and probing their likely mechanisms of action, using both in vitro and in vivo models. In-depth CTC investigation in EC holds the potential to increase knowledge of this disease aggressiveness features and pave the way to new and tailored therapeutic strategies.

## 5. Conclusions

In this study, we convey that both epithelial and stem/mesenchymal CTCs are present in the bloodstream of locally advanced esophageal cancer patients while undergoing therapy. Identified CTCs showed a dynamic change of genomic imbalance during treatment. In addition, CTCs from relapsed patients revealed a higher genomic imbalance compared to CTCs from disease-free patients. Recurrent genomic amplifications were present in CTCs identified after neoadjuvant therapy and in disease free patients compared to those who relapsed. Specific enriched terms emerged from copy number aberration analysis of circulating tumor cells of relapsed patients. Overall, studies on the genomic landscape of CTCs are feasible and provide the unprecedented opportunity to gain an insight into their biology and their perspective clinical significance.

## Figures and Tables

**Figure 1 cancers-13-06369-f001:**
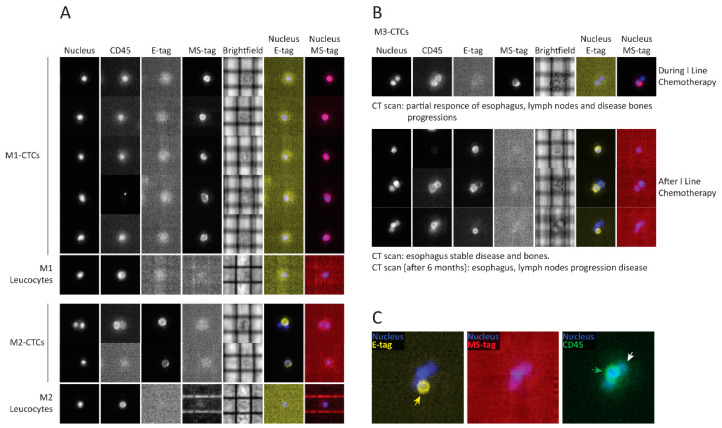
Identification of different CTC phenotypes by Grab-all assay on samples from metastatic esophageal cancer (EC) patients. (**A**) Panel of representative CTCs with different phenotypes, frankly mesenchymal/stem (patient M1) (red pseudocolour), frankly epithelial (patient M2) (yellow pseudocolor); CD45+ matched leukocytes as controls. (**B**) Phenotypic evolution of CTCs of patient M3: single mesenchymal/stem CTC during first line chemotherapy and epithelial CTCs after treatment; CT scan outcomes performed at the time of blood draws or after six months are listed under the panels. (**C**) Zoomed DEPArray image of a cell cluster composed of an epithelial CTC (yellow pseudocolor) and leukocytes (green pseudocolor). Each metastatic EC patient was codified as follows: Mx (patient ID); E-tag: epithelial phenotype tag; MS-tag: mesenchymal-stem phenotype tag.

**Figure 2 cancers-13-06369-f002:**
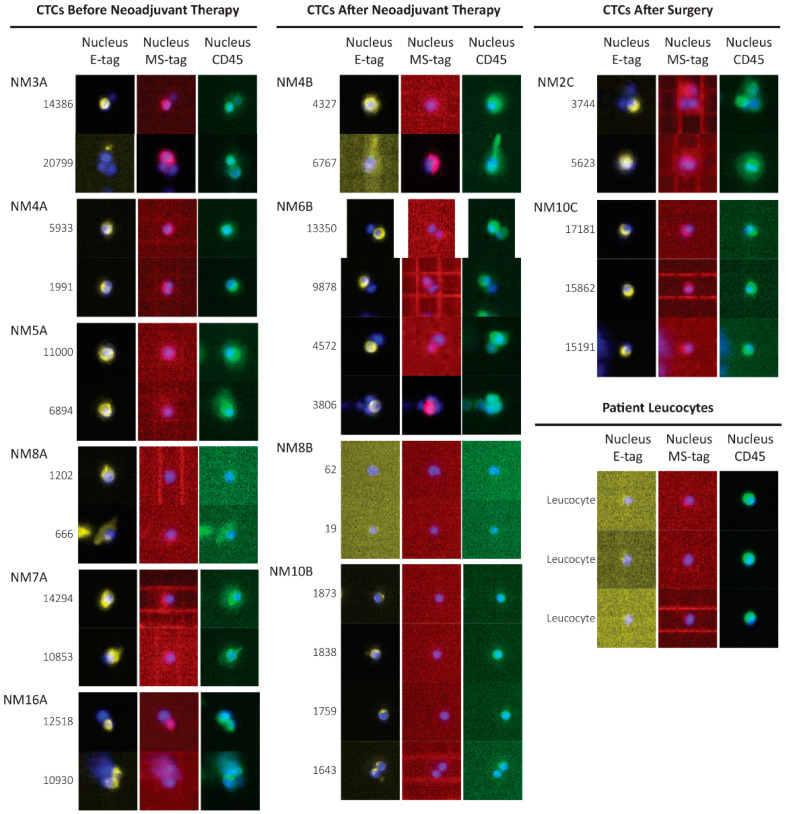
DEPArray panels of representative CTCs identified in non-metastatic esophageal cancer (EC) patients. Images of CTCs obtained before neoadjuvant treatment, after neoadjuvant treatment and after surgery are grouped in separate columns (each including three different imaging channels). A fourth group lists images of patients’ leukocytes. Every line represents an identified cell with its specific ID. Autofluorescence signals on FITC channel (green) of CTCs are well distinct from specific CD45 positivity of leukocytes. We identified frankly epithelial CTCs (yellow pseudocolor) or mesenchymal/stem CTCs (red pseudocolor) or hybrid phenotypes (ID 3806). In patient NM8′s blood drawn after neoadjuvant therapy, we identified cells that were both negative for CD45 and other Grab-all assay tags. We decided to sort these single cells to perform CNA analyses in order to classify them as tumoral or normal cells. The reported numerical codes on the left of each cells represents their DEPArray ID. CTC: circulating tumor cells; CNA: copy number aberration.

**Figure 3 cancers-13-06369-f003:**
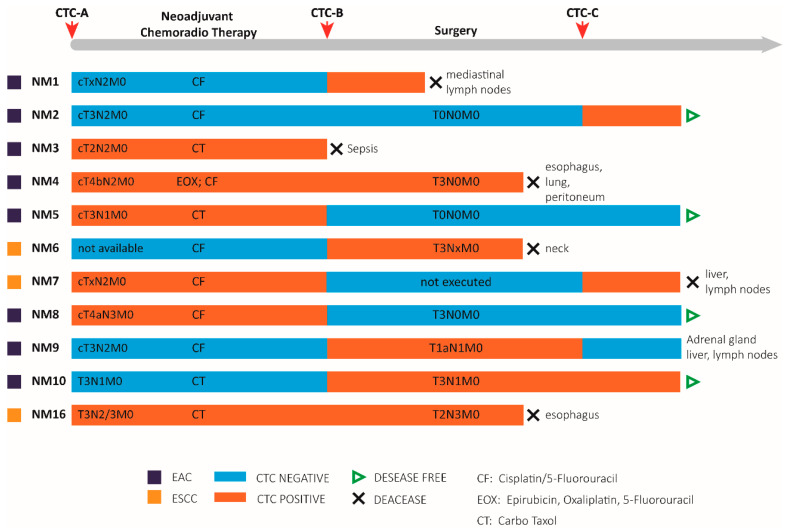
Summary of the clinical outcomes and CTC status of patients with non-metastatic esophageal cancer (EC). For each patient a timeline is defined along which we report: histological types (purple/ochre squares) EAC: esophageal adenocarcinoma cancer; ESCC: esophageal squamous cell carcinoma; TNM clinical classification based on the tumor (T), lymph node (N), and metastasis (M) system. ×: not assessed, neoadjuvant treatments, TNM pathological classification for patients that underwent surgery, clinical status (green arrows indicate disease/free status versus black arrows which indicate death), and in case of disease recurrence the site of relapse was specified. Blood draws time points are represented as red arrows at the top of the gray timeline bar. The presence or absence of CTCs (CTC status) is indicated by the color of the bars (light blue for CTC negativity and red for CTC positivity). CTC: circulating tumor cells.

**Figure 4 cancers-13-06369-f004:**
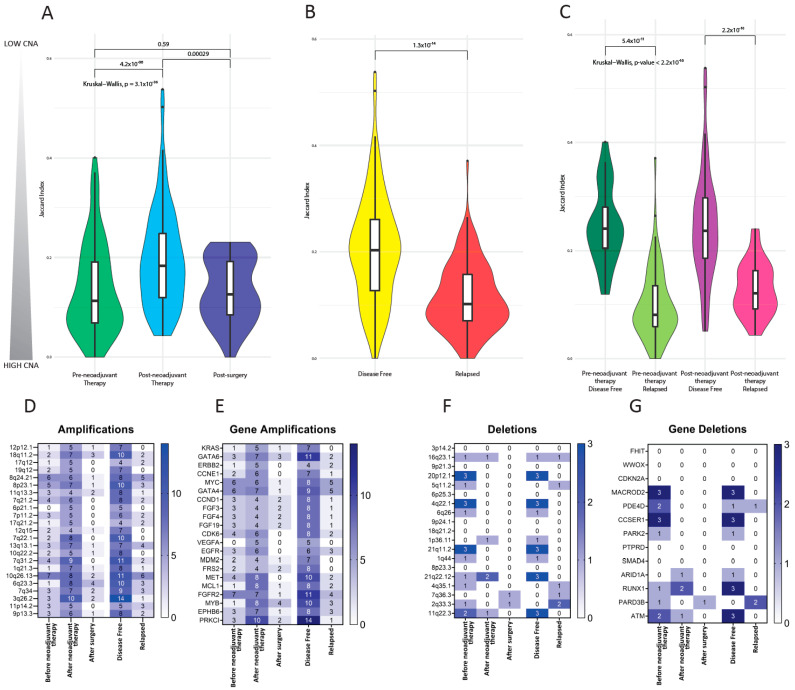
CTC genomic imbalance is measured relative to a matched diploid genome using a Jaccard similarity index (JI) between the set of CNAs present in the CTC and those in the patient’s leukocytes. A high JI value (max 1) indicates that the CTC’s genome has a similar CNA profile to that of matched leukocytes, a small JI value (min 0) indicates a CTC’s genome with a CNA profile that differs from that of the matched leukocytes. JI values close to zero indicate increasing genomic imbalance. (**A**) violin plot representing the distribution of CTC’s JI for cells grouped based on time point of blood draws, pre-neoadjuvant therapy (*n* = 20) (green) post-neoadjuvant therapy (*n* = 27) (blue) post-surgery (*n* = 10) (violet). (**B**) violin plot representing the distribution of CTC’s JI for cells grouped based on patients’ clinical outcomes: disease free (*n* = 28) (yellow) relapsed (*n* = 29) (red). (**C**) violin plot representing the distribution of CTC’s JI for cells grouped based on a combination of time point and patient’s clinical outcomes: pre-neoadjuvant therapy of disease free patients (*n* = 5) (dark green) and of relapsed patients (*n* = 15) (light green); post-neoadjuvant therapy of disease free patients (*n* = 14) (purple) and of relapsed patients (*n* = 13) (pink). At the top, we report the *p*-values of Kruskal–Wallis and Wilcoxon tests. Panel of known esophageal cancer (EC) specific focal copy number changes which we found in identified CTCs. Frequently amplified regions (**D**), genes included within amplified regions (**E**), frequently deleted regions (**F**) and genes included in deleted regions (**G**). CTCs are grouped based on the time point of blood draws and patients’ clinical outcomes. The absolute frequency of CTCs bearing a specified focal copy number change is represented by a number and on a color scale. CTC: circulating tumor cells; CNA: copy number aberration.

**Figure 5 cancers-13-06369-f005:**
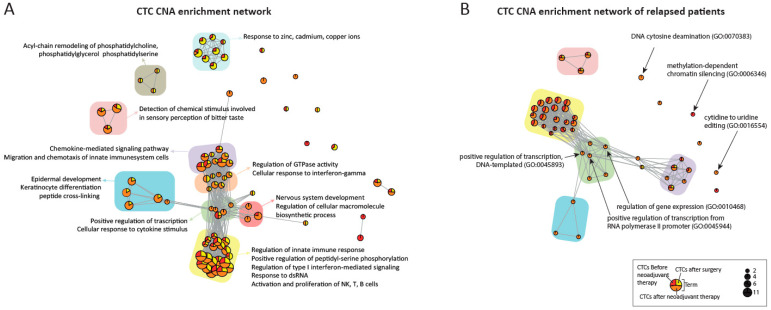
Networks on enriched GO (Biological Process) terms of CNAs of identified CTCs. GO terms are represented as nodes (circles). Sizing of the nodes is based on the number of CTCs with enriched term. Every circle is divided into sectors representing a proportion of the clinical step where CTCs were been identified: red, orange sectors before and after neoadjuvant therapy, respectively, yellow after surgery; edges (grey lines) represent at least ten common genes overlapping the two terms. Network on enriched GO terms of all identified CTCs (**A**) and only of CTCs identified on relapsed patients (**B**). GO terms in the larger communities were color-coded accordingly. Colored rectangles behind nodes: activation and modulation of innate immune system (yellow), chemotaxis and migration of immune system cells (violet), sensory perception of bitter taste (pink), skin development (blue), modulation of the transcription (green), macromolecules biosynthesis/nervous system development (red), response to zinc/cadmium/copper ions (cyan), cellular response to interferon.gamma (orange), acyl-chain remodeling (olive). CTC: circulating tumor cells; GO: Gene Ontology; CNA: copy number aberration.

**Table 1 cancers-13-06369-t001:** CTC counts during patients’ treatment course and clinical outcomes (relapsed or disease free). Blood draws were performed before therapy (A), one month after the end of neoadjuvant therapy (B) and one month after surgery (C). The second and the third blood draws were performed concomitantly to CT scan.

CTC Count					
Before Therapy (A)		After Neoadjuvant Therapy (B)		After Surgery (C)	
20		27		10	
Relapsed	Disease Free	Relapsed	Disease Free	Relapsed	Disease Free
15	5	13	14	1	9

**Table 2 cancers-13-06369-t002:** List of mostly associated genes of the six terms distinctive in CTCs from relapsed patients. CTC: circulating tumor cells; GO: Gene Ontology; EC: esophageal cancer.

Gene	Term	Percentages of CTC in Relapsed Patients (*n* = 29)	Percentages of CTC in Disease-Free Patients (*n* = 28)	Association with EC and Cancer [Reference]
*APOBEC3* *cluster gene*	cytidine to uridine editing (GO:0016554) DNA cytosine deamination (GO:0070383)	24.1%	7.1%	[43,44]
*SALL2*	regulation of gene expression (GO:0010468) positive regulation of transcription, DNA-templated (GO:0045893) positive regulation of transcription from RNA polymerase II promoter (GO:0045944)	27.6%	0%	[45]
*TOX4*	regulation of gene expression (GO:0010468) positive regulation of transcription, DNA-templated (GO:0045893)	27.6%	0%	[46]
*ZNF219*	regulation of gene expression (GO:0010468)	27.6%	0.0%	[47]
*PRMT5*	regulation of gene expression (GO:0010468)	27.6%	3.6%	[48]
*AJUBA*	regulation of gene expression (GO:0010468)	24.1%	3.6%	[49]
*NRL*	regulation of gene expression (GO:0010468) positive regulation of transcription, DNA-templated (GO:0045893) positive regulation of transcription from RNA polymerase II promoter (GO:0045944)	24.1%	3.6%	[50]
*PPARGC1A*	regulation of gene expression (GO:0010468) positive regulation of transcription, DNA-templated (GO:0045893) positive regulation of transcription from RNA polymerase II promoter (GO:0045944)	41.4%	10.7%	[51]
*MAP2K2*	regulation of gene expression (GO:0010468) positive regulation of transcription, DNA-templated (GO:0045893)	27.6%	7.1%	[52]
*BAZ1A*	regulation of gene expression (GO:0010468)	24.1%	7.1%	[53]
*DENND4A*	regulation of gene expression (GO:0010468)	24.1%	7.1%	[54]
*CHD8*	positive regulation of transcription, DNA-templated (GO:0045893) positive regulation of transcription from RNA polymerase II promoter (GO:0045944)	27.6%	0.0%	[55]
*SUPT16H*	positive regulation of transcription, DNA-templated (GO:0045893) positive regulation of transcription from RNA polymerase II promoter (GO:0045944)	27.6%	0.0%	
*CEBPE*	positive regulation of transcription, DNA-templated (GO:0045893) positive regulation of transcription from RNA polymerase II promoter (GO:0045944)	24.1%	3.6%	[56]

## Data Availability

The data that support the findings of this study are available on request from the corresponding author.

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
