# Peer review of "CNA Profiling of Single CTCs in Locally Advanced Esophageal Cancer Patients during Therapy Highlights Unexplored Molecular Pathways"

_cancers, 2021, doi:10.3390/cancers13246369_

Round 1

Reviewer 1 Report

The findings of this study might provide more opportunity to gain an insight for the treatment of esophageal cancer.

Author Response

We thank the reviewer for the comments.

Reviewer 2 Report

Dear Authors,

Thank you very much for addressing the major issues.

Best wishes

Author Response

We thank the reviewer for the comment.

Reviewer 3 Report

Some of the issues I had previously have now been addressed, several have not. The latter still need amendments (please refer to previous numbering):

Please address:

  1. Has this now led to changes in the manuscript text? Is it discussed (as requested) and are these references added? Please provide what lines of the documents these amendments can be found.
  2. Has this now led to changes in the manuscript text and were relevant references added? Please provide what lines of the documents these amendments can be found.
  1. Has this now led to changes in the manuscript text and was the Method section amended? Please provide what lines of the documents these amendments can be found..
  2. Line 267 does not refer to MDA-MB436, please clarify the manuscript text changes for this point and is there a reference included?

8. What amendments of the manuscript text have the authors made, these acronyms need to be introduced in the text. If you want to refer to your previous reference that is fine.

12. Colour coding relevant for a figure needs to be described in the figure legend not the document text.

15. Line 578-579 is your Data availability statement, not a clarification for this issue. Please provide correct location.

Reviewer 4 Report

The authors have made all the proposed changes and no further comments.

Author Response

We thank the reviewer for the comment

This manuscript is a resubmission of an earlier submission. The following is a list of the peer review reports and author responses from that submission.

Round 1

Reviewer 1 Report

Overall:

In this article, the authors confirmed that CTCs (circulating tumor cells) are present in the bloodstream of locally advanced esophageal cancer patients while undergoing therapy. They showed that identified CTCs showed a dynamic change of genomic imbalance during treatment. Moreover, they performed copy number aberration analysis of CTCs of relapsed patients. The findings of this study might provide more opportunity to gain an insight for the treatment of esophageal cancer.

Comments:

1. Are there any different results between esophageal adenocarcinoma cases and esophageal squamous carcinoma cases.

Reviewer 2 Report

Dear Authors,

There are several points  I do not understand:

1)

You wrote that you included a total of 11 patients diagnosed with locally advanced non-metastatic esophageal cancer: 8 (72.7%) adenocarcinomas (EAC), 3 (27.3%) squamous carcinomas. 

How can it be that in Table 1 and Table 2 more patients are described.

For instance Table 1: Relapsed: 15 patients before therapy

Table 2: CTC Relapsed n=29

2)

You wrote: "All patients underwent surgery..."(line 298)

However, later you wrote: "

With the exception of NM1, who relapsed and died after neoadjuvant therapy, the remaining three patients underwent surgery but NM6 died from neck metastases before the third blood biopsy. "

Does that mean that not all patients underwent surgery?

How can someone solely die from neck metastases? Did the metastases led to suffocation? Please explain!

It is technically/ scientifically not correct that someone relapsed after neoadjuvant chemotherapy!

Neoadjuvant Chemotherapy is given to downsize/ shrink a tumor and not to cure it!

So it does not make any sense that someone relapsed and died after neoadjuvant therapy. Please fix the sentence!

3) 

Please explain the abbreviation "GO biological process domain"

Do you mean Gene Ontology? If so, then please explain it!

4)

What are "tableCTCs"? Check line 423!

5)

Please interpret and explain Figure 5?

6)

How many patients did eventually relapse?

7)

You wrote "

Our findings, albeit conducted on a small cohort, demonstrate that EC spreads epithelial as well as mesenchymal CTCs carrying high burden of genomic imbalance, that entails chromosomal aberrations in innate immune system-related loci. "

Your cohort was extremely limited, rather a case series!

So please do not write that your findings demonstrated something. Maybe they indicated something!

8)

The same is true for

"The neoadjuvant therapy has a selective action against CTCs, likely promoting a higher genomic imbalance. "

It MAY have a selective action.

Please, do not exaggerate the importance of your findings and results.

9)

The same goes for:

In conclusion, our work demonstrates for the first time that CTCs of locally advanced EC have a mixed phenotype with high levels of genomic disruption, particularly those CTCs found in relapsed patients. 

It MAY have shown something!

Please avoid sensationalism and be cautious with conclusions!

Reviewer 3 Report

The manuscript by Gallerani et al. is complex and sometimes confusing. There seems to be interesting content mainly that it is implied that broad gene copy number variation may be measurable in EC patient CTCs and correlates to disease state and previous studies on tissue. However, the manuscript falls short of reaching a clarity that allows judgement how reliable that may be the case.

There are a number of issues that I identified:

  1. General: How reliable is it to perform CNA testing after WGA given that WGA is known to have difficulties to evenly amplify the genome? This should be referred to in the discussion.
  2. Lines 83-84 / 236-237: What is the background? What are the studies / references justifying to choose these specific epithelial and mesenchymal markers? What is their relationship to CTCs or as EMT markers in CTCs (previously used/proven or not with references)?
  3. Lines 142: anti ABCG2 would also have been conjugated, I assume with a far red emitter?
  4. Lines 173-177: It is not clear if this test was simply done or if the outcome of this test was used for inclusion/exclusion of samples for further analysis? What was the criteria and what was the drop out %?
  5. Lines 238-239: please provide a reference for MDA-MB436 expressing both epithelial and mesenchymal markers.
  6. Was any staining with these antibodies performed on healthy donor PBMCs to evaluate potential for background staining?
  7. Figure 1 A: A lot of the cells shown to represent CTCs appear clearly positive for CD45 so would not normally defined as CTCs or do I misinterpret this? If not, what is the rationale behind this and do the authors have other data (mutation status or CNA status) that these specific cells are indeed CTCs?
  8. Line 272 what is “yp” and “c”?
  9. Figure 2: is confusing and clear statement what the various acronyms are might help. I assume NM-number are the patients and A, B, C are the timepoints in regard to therapy? Should be included in the legend. What are the other numbers (they also seem to appear in other figures), do they have something to do with the DEPArray method? Would need explaining if included (here and in other figures) or if they are not important in the context they should be deleted as not meaningful for non-DEPArray users. Why is CD45 staining not shown? Some of the leukocytes seem to look very similar to cells that are considered CTCs and that is true for several “CTCs”. Any comment on that? Any other molecular evidence for CTC status?
  10. Figure legend 3 also needs to introduce the acronyms properly starting with the fact that the NM number are the patient samples, I assume. EAC? ESSC? cTxN2M0? TNM? And so forth.
  11. Figure 4: increase size of labels in (A)-(C). It is quite confusing how many CTCs from how many patients would have contributed to each violin plot it seems there were a total of n=53 but I am not clear how many would e.g. be prior treatment, after treatment and after surgery, please add these labels into the violin plots or state them in the legend.
  12. Figure 5: increase font size of in figure labels. There seems to be some confusion what should be outlined in the Figure legend and what is described in the surrounding section 7 Enrichment analysis for instance colour coding in the schematic belongs in the legend.
  13. Line 423 What does “tableCTCs from all time points shared…. “ mean?
  14. I don’t understand Table 2 fully: The reference [column], is for previous report(s) of association with EC, I assume. That should become obvious. The column could be named “Association with EC”, have for each marker a brief statement what the association is with the reference. Currently it appears as if the reference may be for CTC relapsed or disease free. Also not clear what the % values mean. Is this the proportion of n=29 patients or n=28 patients that have CN alterations in these genes or the total proportion of changes?
  15. Lines 474-475 “In fact, post-surgery CTC CNA is similar to that of before therapy CTCs [13].” This sounds as if the finding is either already published or not novel.
  16. The manuscript currently has two conclusion sections one lines 509-518 and an additional one lines 521-530. This could be consolidated into one section.

Reviewer 4 Report

Comments to the Author

The manuscript entitled “CNA profiling of single CTCs in locally advanced esophageal cancer patients during therapy highlights unexplored molecular pathways” is a well-written manuscript .

I have some comments that could possibly be discussed:

Major comments:

The majority of the manuscript is also published in research square (doi: 10.21203/rs.3.rs-850357/v1) from the author's own paper “CNA profiling of single CTCs in locally advanced esophageal cancer patients during therapy highlights unexplored molecular pathways”.

Other comments as follows:

  1. In the introduction, it would be important to highlight the updated cancer statistics that could enrich the paper. Ex: GLOBOCAN 2020 statistics
  2. I would suggest mentioning the study period of patient recruitment.
  3. Tables should be self-explanatory, therefore please add written-out abbreviations to the legends where missing. Ex: Table S1
  4. If using abbreviations for the first time, please state the full term first.

Ex: Line number 409: GO biological process domain

  1. Please check the alignment of ‘term’ in table 2.
  2. There are some typos across the manuscript and I hope that the authors can carefully review the manuscript.

Ex:

  • Line number 60: unnecessary space after “EC”
  • Line number 61: unnecessary space after “CNA”
  1. Concerning the limitations, the authors could highlight the limitations of the study.
  2. It is observed that most of the reference (Ex:5,8,16,17,24,) is incomplete.

Ex: In Bibliography,

  • Reference No: 5,8,16,17 - Page number is missing.